# Sustaining the Benefits of Social Media on Users' Health Beliefs Regarding COVID-19 Prevention

**Huan-Ming Chuang and Yi-Deng Liao \*** 

Department of Information Management, National Yunlin University of Science and Technology, Yunlin 640, Taiwan; chuanghm@yuntech.edu.tw
\* Correspondence: d10823001@yuntech.edu.tw or a0975918358@gmail.com; Tel.: +886-975-918-358

**Abstract:** During the COVID-19 pandemic, social media has facilitated the efficient and effective dissemination of healthcare information and helped governments keep in touch with their citizens. Research has indicated that social media can exert negative and positive influences on users' mental health. One negative effect is social media fatigue caused by information overload. However, under the current pandemic, comprehensive research has yet to be executed on the effect exerted by social media on users' health beliefs and subjective well-being (SWB). Consequently, we conducted our research to probe the influence of social media on users' perceptions of COVID-19 prevention. This study established a research model based on 340 valid responses to an online questionnaire survey from Taiwan. SmartPLS 3.0 was used to verify the developed measurement and structural models. We found social media users' incidental and focused knowledge gain positively related to their social media intensity. In addition, social media intensity positively correlated with health beliefs and SWB. Accordingly, we can determine that proper social media use can enhance health beliefs. Based on our derived findings, we propose a set of practical recommendations to leverage social media effectively and sustainably during, and after, the COVID-19 pandemic.

**Keywords:** COVID-19; health belief model; social media fatigue; subjective well-being

## 1. Introduction

COVID-19 inherently spreads rapidly and engenders severe effects on the human respiratory system [1,2]. Ezeah, Ogechi, Ohia, and Celestine [3] stated that the global spread of this virus has shown that health systems worldwide must be reorganized entirely to improve the effectiveness of their responses to emergencies in public health. During the pandemic, to curb the virus's spread, many countries worldwide have closed down their economies, suspended international flights, and placed strict restrictions on the movement of people in an attempt to restrict the spread. Currently, vaccines constitute the primary tool for limiting this virus's spread and preventing severe symptoms.

The COVID-19 pandemic seriously threatens the sustainability of human mental health or subjective well-being. As specified by the World Health Organization (WHO), "mental health is a state of well-being in which an individual is aware of his or her capabilities, able to cope with normal life stresses, work productively, and contribute to his or her community". Due to its capability of affecting physical health, relationships, and daily life, a positive state of mental health is vital for enabling people to transition from fear regarding the effects of a pandemic to their original life patterns.

To overcome the fear of the COVID-19 pandemic, preventive measures based on proper health behaviors have been critical. These behaviors relate to those actions that support people's health and prevent their vulnerability to viruses. The virus, as mentioned above, inherently does not travel, as confirmed by the WHO; instead, individuals carry it. Consequently, the WHO emphasized that adopting appropriate health behaviors is critical for controlling the pandemic. The health behaviors suggested by the WHO for

preventing the spread of COVID-19 include using hand sanitizers, washing hands frequently, maintaining personal hygiene, using protective materials such as face masks, practicing social distancing, and staying at home. The behaviors mentioned above have been widely promoted on various social media platforms, and such platforms have also doubled as the primary source of information related to COVID-19 [4]. As a result, the changes caused by the pandemic in people's social media usage are worthy of investigation. Specifically, Sheth [5] claimed that social media benefits knowledge acquisition [6]; therefore, the knowledge gain patterns and social media intensity during the pandemic are primary concerns.

Furthermore, since knowledge gain is an essential method for controlling the COVID-19 pandemic, its impact on users' health beliefs, such as perceived threats and expectations, has not been investigated thoroughly. Besides this, studies on the effects of social media have focused either on users' social media fatigue or their subjective well-being (SWB); a holistic perspective can gain deeper insights into this issue.

In summary, to bridge these gaps, under the context of the COVID-19 pandemic, this study addresses the following research questions:

RQ1. What are the patterns of knowledge gain and their impact on social media intensity?

RQ2. What effect does social media intensity have on users' health beliefs regarding perceived threats and expectations?

RQ3. How to sustain the benefits of social media effectively, considering social media fatigue and subjective well-being?

These research questions motivate us to pursue the goal of leveraging the benefits of social media to counter-attack the COVID-19 pandemic and further sustain human subjective well-being.

## 2. Theoretical Background

We established our study framework based on the stimulus–organism–response (S–O–R) model. The S–O–R model was developed in a previous study [7] based on concepts related to environmental psychology; in this model, all aspects of the environment are considered to be external stimuli that affect individuals' internal cognition and emotions and thus trigger the individuals' behavioral responses. For the current study, knowledge gain in social media was identified as the stimulus; social media intensity was identified as the organism; and health beliefs, social media fatigue, and SWB were identified as responses.

### 2.1. Knowledge Gain on Social Media

Knowledge acquisition is a social media-derived benefit. Barker, Dozier, Weiss, and Borden [8] recognized two types of knowledge gain on social media. First, focused knowledge gain occurs when users search for information or learn the knowledge they desire. Second, incidental knowledge gain occurs when users search for information on social media and accidentally obtain additional information or learn new knowledge.

### 2.2. Social Media Intensity

Social media intensity refers to a user's activity level and engagement with social media. It significantly affects user social capital establishment and maintenance. Research suggested the use of Facebook to be strongly associated with the following forms of social capital: bridging, bonding, and maintained social capital [9]. In addition, social capital enables individuals to leverage resources, such as the ability to organize groups, personal relationships, and useful information [10] from others.

### 2.3. Health Belief Model

Constructed in the 1950s, the health belief model (HBM) elucidates individuals' failure to participate in disease prevention and detection [11–13]. This model is among those frequently applied for understanding individuals' health behavior [14–16] in the psychosocial field. Behavioral beliefs in this model can be partitioned into two primary classes, one of

which is perceived expectation and the other is perceived threat. Perceived threat refers to the psychological feeling experienced by people when they are in trouble or their lives are threatened. It can be evaluated using two factors: perceived severity and perceived susceptibility. Furthermore, perceived expectation refers to the perceived benefits related to how effective specific actions minimize the risk of health problems [17]. It also includes the perceived barriers that individuals believe may inhibit their execution of particular health behaviors [18,19].

### 2.3.1. Perceived Susceptibility

Perceived susceptibility is a person's own feeling about their chances of contracting a disease [20,21]; it represents how they believe they are susceptible to infection [22]. For instance, people make efforts to protect themselves against COVID-19 (e.g., by adopting behaviors such as following the health-related instructions of relevant authorities) only if they believe that a possibility exists of them being infected with COVID-19.

### 2.3.2. Perceived Severity

In addition, perceived severity is a person's appraisal of a disease's possible consequences from a medical (e.g., death, disability, or pain) and social (e.g., effects on social relations, work, or family life) perspective [20]. It indicates how people are concerned about a disease's medical and social implications [22]. For example, the seriousness of COVID-19 is well known. The COVID-19 infection risk is true for everyone, considering confirmed cases worldwide, regardless of race, gender, or age. COVID-19 infection can cause severe lung damage, which is called pulmonary fibrosis, and might result in death. In addition, even those who are not infected with COVID-19 have still been affected by the pandemic. Governments have restricted the movement of people, which has affected the economies of countries, and has resulted in many people losing their jobs. Accordingly, this pandemic has had a severe influence on all sections of society.

### 2.3.3. Perceived Benefits

Perceived benefits are a person's feelings about the effectiveness of various actions adopted to counteract a disease threat [20]. For instance, to protect themselves against COVID-19, people have engaged in pandemic prevention behaviors, such as obtaining relevant information about the pandemic and following the instructions of relevant authorities.

### 2.3.4. Perceived Barriers

Moreover, perceived barriers are a person's feelings about the possible negative aspects of a specific health-related action. Individuals often conduct a cost–benefit analysis to appraise a health action's positive aspects against its negative aspects, including associated iatrogenic effects, side effects, inconvenience, unpleasant experiences, or time-consumption execution of the proposed action [20]. In the case of the COVID-19 pandemic, daily pandemic prevention activities, such as checking body temperature, washing hands, wearing a mask, maintaining social distance, and confirming case footprint, are time-consuming and inconvenient. Moreover, individuals can experience unpleasantness and anger when they are aware of pandemic-related information but see others not following appropriate disease prevention behavior.

### *2.4. Social Media Fatigue and SWB*
### 2.4.1. Social Media Fatigue

Fatigue occurs when people engage in activities that require energy (both physical and psychological) consumption and motivation over a long time. Usually, when people perform an action whose cognitive and physical load exceed those they can handle, they experience pressure, weakness, and eventually physical or psychological fatigue. Two types of fatigue have been identified: psychological and physical fatigue. Psychological fatigue refers to adverse cognitive conditions that lead to pathological stress, boredom,

or anxiety. Physical fatigue affects individuals' physiological state and causes them to experience conditions such as physical imbalance, eye fatigue, or muscle fatigue [23].

In social media, the corresponding fatigue in users can be determined to be a subjective multidimensional experience that includes adverse emotional reactions (e.g., disappointment, anger, diminished motivation to use social media, loss of interest, or vigilance; [24]). Social media fatigue occurs when people become tired of social media content or have an excessive number of friends and contacts on social media [25]. It manifests as information overload and social overload that make users feel stressed and tired [26].

### 2.4.2. SWB

We can also define SWB as a person's affective and cognitive assessments of their overall life. Ideal SWB is generally linked to a high level of life satisfaction and pleasant emotions and a low level of negative emotions [27].

### 3. Research Model and Hypotheses Development

Based on the S–O–R model, this study considered focused and incidental knowledge gain as stimuli; social media intensity as an organism; health beliefs, social media fatigue, and SWB as user responses.

Considering that knowledge is power in efforts to curb the COVID-19 pandemic, we propose that focused and incidental knowledge gain enhances social media intensity. Social media intensity further exerts two-fold influences. First, it influences health beliefs during an unusual period. Second, it affects people's perception of SWB and social media fatigue. Figure 1 illustrates our established research framework.

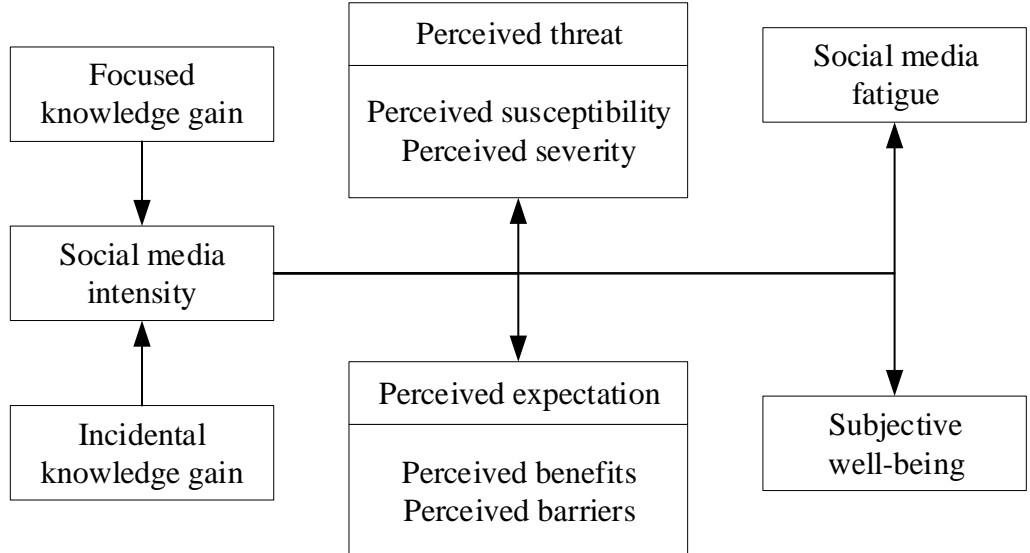

**Figure 1.** Research Framework.

### 3.1. Influence of Knowledge Gain on Social Media Intensity

Self-construal and materialism strongly influence social media intensity [28]. In the current pandemic, interest in health-related knowledge has increased among social media users. Accordingly, this study used the protection motivation theory [29] to predict social media users' intention to protect themselves under the fear of threats, particularly health-related threats. Knowledge gain was identified as a significant motivation for individuals to defend themselves against COVID-19. The convenience and instantaneousness of social media enable users to search for desired information and knowledge regarding COVID-19. Therefore, this study postulated that people's knowledge gain behavior would increase during a pandemic. Consequently, we drew up the following hypotheses:

**Hypothesis H1a.** *Focused knowledge gain and social media intensity would be positively related.*

**Hypothesis H1b.** *Incidental knowledge gain and social media intensity would be positively related.*

*3.2. Influence of Social Media Intensity on Health Beliefs*

Before the widespread availability of COVID-19 vaccines, people had to protect themselves from COVID-19 through preventive health behaviors, and they required sufficient health-related knowledge to engage in these behaviors. Although television news, newspapers, and magazines convey information about the pandemic, they are less immediate relative to social media. Therefore, people have often relied on social media to obtain first-hand knowledge in the fight against COVID-19. Thus, social media intensity could be expected to be associated with social media users' health-related perceptions, including perceived barriers, severity, benefits, and susceptibility. Accordingly, we drew up the ensuing hypotheses:

**Hypothesis H2a.** *Social media intensity and social media users' perceived susceptibility would be positively related.*

**Hypothesis H2b.** *Social media intensity and social media users' perceived severity would be positively related.*

**Hypothesis H2c.** *Social media intensity and social media users' perceived benefits would be positively related.*

**Hypothesis H2d.** *Social media intensity and social media users' perceived barriers would be positively related.*

*3.3. Influences Exerted by Social Media Intensity on Social Media Fatigue and SWB*

Social media fatigue might occur because of inappropriate social media usage. For example, the extension of repetitive tasks engenders diminished task performance [30]. The convenience and instantaneousness of social media enable users to search for desired information and knowledge easily. Information overload occurs when a person has insufficient cognitive ability or time to accommodate or process information [31]. In addition, information overload causes social media fatigue, as indicated by Cao and Sun [26]. Accordingly, we drew up the following hypothesis:

**Hypothesis H3a.** *Social media intensity and social media users' social media fatigue would be positively related.*

Social media is multifunctional. For instance, users can establish their social status on social media, connect with friends to maintain social relationships, share their knowledge and experiences with others, and gain desired knowledge. These functions provide numerous opportunities for psychological satisfaction, contributing to people's SWB. Therefore, we drew up the following hypothesis:

**Hypothesis H3b.** *Social media intensity and social media users' SWB would be positively related.*

**4. Methodology**

Our data collection approach entailed administering a survey to social media users in Taiwan. The data collection, measurement, and analysis procedures are described in the following text.

*4.1. Questionnaire Design*

The first part of the adopted questionnaire was related to the basic information of the respondent, and the second part was related to the primary research constructs. Question-

naire items used a 7-point Likert scale with anchors ranging from 1, representing "strongly disagree", to 7, meaning "strongly agree".

Measurement items were adopted from the existing literature with some wording modifications to ensure content validity and its fit to our research contexts. These items were originally in English and translated into Chinese. This study further recruited five domain experts for a pretest to ensure the translation was accurate and easy to understand. Then the revised questionnaire was distributed to 50 suitable samples for a pilot test to confirm the appropriateness of the questionnaire items before wide distribution.

### 4.2. Sample and Data Collection

The questionnaire was distributed over the Internet through convenience and snowball sampling to individuals with experience using Facebook. Data were collected from 25 May to 7 June 2021, when Taiwan was at level three COVID-19 alert, due to situations of confirmed cases. A total of 340 valid questionnaires were collected. Since data were collected around two weeks, possible early, versus late, respondent bias was not an issue. Besides, online questionnaires effectively overcame the problem of missing data. The collected data, IBM SPSS 22.0 was used for descriptive statistics of the sample profile. Moreover, SmartPLS 3.0 was conducted to perform structural equation modeling (SEM) analysis.

### 4.3. Measurement Items

We adapted measurement items for our executed study from the literature. These are tabulated in the following table (Table 1).

**Table 1.** Measurement items.

| Construct (Source) | Explanatory Items |
|---|---|
| Focused knowledge gain [8] | 1. I often learn something I need to know when visiting this site.<br>2. This site effectively communicates what I need to know.<br>3. This site helps me learn what I need to know. |
| Incidental knowledge gain [8] | 1. I enjoy learning new things by accident when visiting this site.<br>2. I often learn interesting things that I was not looking for when visiting this site.<br>3. Sometimes, I learn something new that was not intended when visiting this site.<br>4. When visiting this site, I sometimes get a bit distracted by new information I was not looking for. |
| Social media intensity [8] | 1. This site is a part of my everyday activity.<br>2. I feel out of touch when I have not logged into this site for a while.<br>3. I would feel sorry if this site shut down. |
| Perceived susceptibility [20] | 1. I feel that I have a high chance of catching COVID-19 infection during the pandemic.<br>2. I feel that my physical health increases my likelihood of catching COVID-19 infection during the pandemic.<br>3. I feel that I have a high chance of catching COVID-19 infection in the future.<br>4. I feel that I have a big probability than others of catching COVID-19 infection during the pandemic.<br>5. I am considerably concerned about catching COVID-19 infection during the pandemic.<br>6. I will be infected with COVID-19 during the pandemic. |
| Perceived severity [20] | 1. The thought of COVID-19 scares me during the pandemic.<br>2. Thinking about COVID-19 makes me feel nauseous during the pandemic.<br>3. Catching COVID-19 infection during the pandemic would endanger my career.<br>4. Thinking about COVID-19 during the pandemic makes my heart beat faster.<br>5. Catching COVID-19 infection during the pandemic would endanger my marriage (or significant relationships).<br>6. COVID-19 is a hopeless disease.<br>7. My feelings about myself would change if I become infected with COVID-19 during the pandemic.<br>8. I am afraid to even think about COVID-19 during the pandemic.<br>9. Catching COVID-19 infection during the pandemic would endanger my financial security.<br>10. The problems that I would experience from COVID-19 would last a long time.<br>11. Infection with COVID-19 causes more severe symptoms than does infection with other diseases.<br>12. Catching COVID-19 infection during the pandemic would change my entire life. |
| Perceived benefits [20] | 1. Gaining information about COVID-19 can prevent future problems during the pandemic.<br>2. I have a lot to gain by acquiring information about COVID-19 during the pandemic.<br>3. Gaining information about COVID-19 can help me avoid COVID-19 infection during the pandemic.<br>4. If I gain information about COVID-19, I might be able to detect a symptom before disease screening during the pandemic.<br>5. I would not be so anxious about COVID-19 If I had gained information about COVID-19 during the pandemic. |
| Perceived barriers [20] | 1. Gaining information about COVID-19 has bothered me during the pandemic.<br>2. To gain information about COVID-19, I had to give up quite a bit during the pandemic.<br>3. Gaining information about COVID-19 can be painful during the pandemic.<br>4. Gaining information about COVID-19 is a time-consuming process.<br>5. My family or friends would make fun of me if I learned about COVID-19 during the pandemic.<br>6. The practice of obtaining information about COVID-19 interferes with my activities during the pandemic.<br>7. Gaining information about COVID-19 would require starting a new habit, which is difficult during the pandemic.<br>8. I am afraid I would not gain information about COVID-19 during the pandemic. |

**Table 1.** *Cont.*

| Construct (Source) | Explanatory Items |
|---|---|
| Social media fatigue [32,33] | 1. Sometimes, I feel tired when using Facebook.<br>2. Sometimes, I feel bored when using Facebook.<br>3. Sometimes, I feel drained when using Facebook.<br>4. Sometimes, I feel worn out from using Facebook.<br>5. I feel disinterested in whether new things are occurring on Facebook.<br>6. I feel indifferent about reminders or alerts of new things from Facebook. |
| Subjective well-being [34] | 1. My online social life on Facebook is close to ideal in most respects.<br>2. The conditions of my online social life on Facebook are excellent.<br>3. I am satisfied with my online social life on Facebook.<br>4. So far, I have obtained the critical things that I want from my online social life on Facebook. |

## 5. Results

### 5.1. Descriptive Statistics

We list in Table 2 the descriptive statistics that we derived for the collected responses.

**Table 2.** Profile of the research sample.

| Category | Features | Number | Percentage |
|---|---|---|---|
| Gender | Male | 172 | 51% |
| | Female | 168 | 49% |
| Age | Under the age of 20 | 22 | 6% |
| | 21–25 | 114 | 34% |
| | 26–30 | 41 | 12% |
| | 31–35 | 45 | 13% |
| | 36–40 | 42 | 12% |
| | Above the age of 41 | 76 | 22% |
| Education | Elementary school | 2 | 1% |
| | Junior high school | 1 | 0% |
| | Five-year junior college program | 17 | 5% |
| | Senior high school | 39 | 11% |
| | Associate degree | 8 | 2% |
| | Bachelor Degree | 183 | 54% |
| | Master degree | 82 | 24% |
| | Ph.D. degree | 8 | 2% |
| Average daily usage of FB | Less than 30 min | 22 | 6% |
| | 30 min to less than 1 h | 75 | 22% |
| | 1 h to less than 2 h | 92 | 27% |
| | 2 h and more | 151 | 44% |

### 5.2. Measurement Model

We evaluated our established measurement model's internal consistency reliability by applying the following measures [35]: composite reliability (CR) and Cronbach's alpha. In addition, we appraised our established model's convergent validity by deriving the average variance extracted (AVE) and outer loadings. Finally, we derived our established model's discriminative validity by employing Fornell–Larcker criteria and cross-loadings.

#### 5.2.1. Internal Consistency Reliability

If, in general circumstances, a model's Cronbach's alpha value exceeds 0.7, then the model can be deemed to exhibit acceptable internal consistency. Since all indicators

are assumed to have the same outer loading when calculating Cronbach's alpha, which is different from the assumption in partial least squares–SEM, the CR is preferred for determining the internal consistency reliability. Cronbach's alpha is a more conservative criterion than the CR to determine the internal consistency reliability. The results obtained for the factors mentioned above in this study are summarized in Table 3. As all the values obtained for these factors are higher than 0.7, the measurement model had satisfactory internal consistency reliability.

**Table 3.** Internal consistency reliability of the research constructs in terms of the Cronbach's alpha and CR.

| Construct | Cronbach's Alpha | Composite Reliability (CR) |
|---|---|---|
| Focused knowledge gain (FKG) | 0.922 | 0.951 |
| Incidental knowledge gain (IKG) | 0.871 | 0.911 |
| Social media intensity (SMI) | 0.848 | 0.908 |
| Perceived susceptibility (PS) | 0.871 | 0.902 |
| Perceived severity (PSV) | 0.91 | 0.924 |
| Perceived benefits (PB) | 0.913 | 0.935 |
| Perceived barriers (PBR) | 0.875 | 0.9 |
| Social media fatigue (SMF) | 0.856 | 0.883 |
| Subjective well-being (SWB) | 0.886 | 0.921 |

### 5.2.2. Convergent Validity

The extent of the correlation between different measurement variables (or indicators) of the same factor can be derived through the convergent validity measure [35]. The outer loading usually measures the convergent validity of the reflecting factors, and the square of the outer loading is also called the indicator reliability. The overall mean of the indicator reliability of each factor is the AVE. Satisfactory convergent validity is indicated by an outer loading of ≥0.7, an indicator reliability of ≥0.5, and an AVE of ≥0.5. The values obtained for the outer loading and AVE in this study are presented in Table 4, and these values indicate that the research constructs had satisfactory convergent validity.

**Table 4.** Convergent validity of the research constructs with regard to the outer loadings and AVE.

| | PS | PSV | PB | PBR | SMI | FKG | IKG | SMF | SWB |
|---|---|---|---|---|---|---|---|---|---|
| AVE | 0.608 | 0.506 | 0.742 | 0.532 | 0.766 | 0.865 | 0.721 | 0.562 | 0.744 |
| FKG1 | 0.168 | 0.319 | 0.257 | 0.208 | 0.638 | 0.916 | 0.722 | −0.201 | 0.497 |
| FKG2 | 0.161 | 0.356 | 0.252 | 0.185 | 0.604 | 0.929 | 0.751 | −0.231 | 0.604 |
| FKG3 | 0.159 | 0.334 | 0.271 | 0.153 | 0.599 | 0.946 | 0.74 | −0.246 | 0.597 |
| IKG1 | 0.143 | 0.326 | 0.22 | 0.183 | 0.593 | 0.749 | 0.863 | −0.194 | 0.564 |
| IKG2 | 0.149 | 0.307 | 0.271 | 0.058 | 0.519 | 0.676 | 0.881 | −0.201 | 0.492 |
| IKG3 | 0.107 | 0.3 | 0.314 | 0.074 | 0.561 | 0.726 | 0.9 | −0.166 | 0.552 |
| IKG4 | 0.137 | 0.27 | 0.245 | 0.167 | 0.388 | 0.502 | 0.744 | 0.07 | 0.365 |
| SMI1 | 0.134 | 0.285 | 0.265 | 0.164 | 0.879 | 0.6 | 0.54 | −0.192 | 0.474 |
| SMI2 | 0.208 | 0.33 | 0.104 | 0.336 | 0.884 | 0.562 | 0.517 | −0.167 | 0.412 |
| SMI3 | 0.173 | 0.379 | 0.22 | 0.212 | 0.863 | 0.573 | 0.561 | −0.158 | 0.478 |
| PS1 | 0.69 | 0.237 | 0.098 | 0.183 | 0.146 | 0.101 | 0.082 | 0.134 | 0.057 |
| PS2 | 0.813 | 0.425 | 0.117 | 0.201 | 0.188 | 0.148 | 0.111 | 0.131 | 0.135 |
| PS3 | 0.854 | 0.413 | 0.005 | 0.22 | 0.171 | 0.18 | 0.133 | 0.086 | 0.165 |
| PS4 | 0.851 | 0.407 | 0.037 | 0.233 | 0.148 | 0.151 | 0.141 | 0.131 | 0.143 |

|        | PS    | PSV   | PB     | PBR    | SMI    | FKG    | IKG    | SMF    | SWB    |
|--------|-------|-------|--------|--------|--------|--------|--------|--------|--------|
| PS5    | 0.777 | 0.577 | 0.187  | 0.213  | 0.147  | 0.137  | 0.141  | 0.132  | 0.16   |
| PS6    | 0.671 | 0.418 | 0.131  | 0.147  | 0.082  | 0.068  | 0.14   | 0.161  | 0.12   |
| PSV1   | 0.543 | 0.711 | 0.235  | 0.24   | 0.275  | 0.205  | 0.21   | 0.06   | 0.207  |
| PSV2   | 0.491 | 0.78  | 0.233  | 0.341  | 0.278  | 0.293  | 0.242  | 0.031  | 0.214  |
| PSV3   | 0.404 | 0.762 | 0.254  | 0.335  | 0.269  | 0.23   | 0.26   | 0.035  | 0.221  |
| PSV4   | 0.414 | 0.73  | 0.174  | 0.457  | 0.283  | 0.32   | 0.239  | −0.04  | 0.23   |
| PSV5   | 0.401 | 0.719 | 0.211  | 0.301  | 0.309  | 0.273  | 0.312  | −0.016 | 0.289  |
| PSV6   | 0.335 | 0.776 | 0.134  | 0.344  | 0.312  | 0.257  | 0.217  | 0.032  | 0.222  |
| PSV7   | 0.317 | 0.686 | 0.252  | 0.191  | 0.257  | 0.224  | 0.228  | 0.094  | 0.165  |
| PSV8   | 0.474 | 0.811 | 0.186  | 0.319  | 0.312  | 0.286  | 0.262  | 0.054  | 0.245  |
| PSV9   | 0.25  | 0.673 | 0.233  | 0.21   | 0.215  | 0.225  | 0.296  | 0.111  | 0.215  |
| PSV10  | 0.274 | 0.579 | 0.436  | 0.014  | 0.199  | 0.246  | 0.268  | 0.085  | 0.218  |
| PSV11  | 0.267 | 0.59  | 0.185  | 0.216  | 0.222  | 0.223  | 0.217  | −0.059 | 0.212  |
| PSV12  | 0.239 | 0.678 | 0.181  | 0.273  | 0.268  | 0.294  | 0.301  | 0.016  | 0.254  |
| PB1    | 0.089 | 0.221 | 0.855  | −0.051 | 0.192  | 0.22   | 0.239  | 0.03   | 0.172  |
| PB2    | 0.116 | 0.281 | 0.906  | −0.101 | 0.228  | 0.276  | 0.307  | 0.037  | 0.221  |
| PB3    | 0.058 | 0.239 | 0.867  | −0.084 | 0.14   | 0.186  | 0.208  | 0.092  | 0.149  |
| PB4    | 0.108 | 0.274 | 0.882  | −0.104 | 0.191  | 0.222  | 0.267  | 0.055  | 0.168  |
| PB5    | 0.114 | 0.296 | 0.792  | 0.052  | 0.2    | 0.276  | 0.279  | 0.016  | 0.226  |
| PBR1   | 0.163 | 0.226 | −0.219 | 0.755  | 0.152  | 0.136  | 0.076  | 0.06   | 0.221  |
| PBR2   | 0.116 | 0.28  | −0.045 | 0.792  | 0.239  | 0.182  | 0.102  | 0.033  | 0.256  |
| PBR3   | 0.139 | 0.218 | −0.18  | 0.837  | 0.135  | 0.116  | 0.061  | 0.182  | 0.105  |
| PBR4   | 0.138 | 0.238 | −0.034 | 0.825  | 0.211  | 0.149  | 0.081  | 0.089  | 0.142  |
| PBR5   | 0.213 | 0.202 | −0.221 | 0.635  | 0.156  | 0.086  | 0.058  | 0.228  | 0.085  |
| PBR6   | 0.154 | 0.182 | −0.154 | 0.63   | 0.064  | −0.009 | 0.016  | 0.247  | 0.065  |
| PBR7   | 0.207 | 0.346 | 0.151  | 0.576  | 0.18   | 0.157  | 0.172  | 0.129  | 0.137  |
| PBR8   | 0.319 | 0.438 | 0.086  | 0.74   | 0.278  | 0.186  | 0.153  | 0.159  | 0.126  |
| SMF1   | 0.163 | 0.087 | 0.007  | 0.289  | 0.025  | 0.014  | 0.038  | 0.637  | −0.098 |
| SMF2   | 0.135 | 0.034 | 0.05   | 0.176  | −0.173 | −0.264 | −0.177 | 0.852  | −0.235 |
| SMF3   | 0.13  | 0.033 | 0.007  | 0.209  | −0.167 | −0.189 | −0.13  | 0.853  | −0.232 |
| SMF4   | 0.204 | 0.111 | −0.001 | 0.31   | −0.047 | −0.067 | −0.041 | 0.746  | −0.088 |
| SMF5   | 0.144 | 0.076 | −0.022 | 0.153  | −0.075 | −0.077 | −0.031 | 0.587  | −0.01  |
| SMF6   | 0.088 | 0.004 | 0.096  | −0.016 | −0.165 | −0.176 | −0.119 | 0.741  | −0.119 |
| SWB1   | 0.193 | 0.3   | 0.136  | 0.191  | 0.439  | 0.442  | 0.448  | −0.137 | 0.848  |
| SWB2   | 0.154 | 0.249 | 0.178  | 0.158  | 0.463  | 0.546  | 0.49   | −0.212 | 0.893  |
| SWB3   | 0.1   | 0.218 | 0.257  | 0.129  | 0.393  | 0.497  | 0.497  | −0.206 | 0.864  |
| SWB4   | 0.127 | 0.314 | 0.2    | 0.229  | 0.489  | 0.598  | 0.593  | −0.182 | 0.846  |

### 5.2.3. Discriminant Validity

Discriminant validity ascertains how a factor is different from other factors in a model, representing the uniqueness of a factor in describing an aspect that is not explained by the other factors. The cross-loading test is a general method for testing the discriminant validity of a construct. The outer loading of each index of a factor should be higher than that of other factors. We summarize in Table 4 the outer loading of each factor in this study; as displayed in this table, the constructs had satisfactory discriminant validity. Furthermore, the Fornell–Larcker criterion can be applied to test a factor's discriminant validity. It

stipulates that for a particular factor to be considered to exhibit discriminant validity, the square root of the AVE (which is derived for this factor) should exceed the coefficients of this factor's correlations with other factors in the model. The results obtained for the Fornell–Larcker criterion are presented in Table 5, and these results indicate that the factors had satisfactory discriminant validity.

**Table 5.** Discriminant validity results for the research constructs in terms of the Fornell–Larcker criterion.

|  | FKG | IKG | SMI | PS | PSV | PB | PBR | SMF | SWB |
|---|---|---|---|---|---|---|---|---|---|
| FKG | 0.711 | | | | | | | | |
| IKG | 0.048 | 0.75 | | | | | | | |
| SMI | 0.164 | 0.524 | 0.779 | | | | | | |
| PS | 0.362 | −0.24 | 0.175 | 0.93 | | | | | |
| PSV | 0.355 | −0.162 | 0.156 | 0.794 | 0.849 | | | | |
| PB | 0.307 | 0.048 | 0.116 | 0.28 | 0.307 | 0.861 | | | |
| PBR | 0.392 | 0.188 | 0.259 | 0.195 | 0.138 | −0.066 | 0.73 | | |
| SMF | 0.379 | −0.188 | 0.196 | 0.659 | 0.617 | 0.226 | 0.27 | 0.875 | |
| SWB | 0.314 | −0.218 | 0.164 | 0.613 | 0.595 | 0.225 | 0.207 | 0.519 | 0.863 |

### 5.2.4. Common Method Bias

Due to the fact that the same respondent constituted the data source for all measurement items, including dependent variable-related and independent variable-related items, common method bias (CMB) might have occurred in this study. According to Podsakoff and Organ [36], CMB can be tested through Harman's single-factor analysis. CMB exists when one factor explains the majority of the variation in exploratory factor analysis. Table 6 presents the results of principal component analysis obtained in this study. We determined that CMB did not constitute a problem for our study because the first factor explained 41.389% of the varince in our exploratory factor analysis.

**Table 6.** Results of principal component analysis.

| Factor | Initial Eigenvalues | | |
|---|---|---|---|
|  | Total | % of Variance | Cumulative % |
| 1 | 12.039 | 23.606 | 23.606 |
| 2 | 6.245 | 12.244 | 35.850 |
| 3 | 4.523 | 8.870 | 44.720 |
| 4 | 3.288 | 6.447 | 51.167 |
| 5 | 2.479 | 4.860 | 56.028 |
| 6 | 1.862 | 3.650 | 59.678 |
| 7 | 1.590 | 3.118 | 62.796 |
| 8 | 1.217 | 2.387 | 65.183 |
| 9 | 1.133 | 2.221 | 67.404 |
| 10 | 1.018 | 1.996 | 69.400 |

### 5.3. Structural Model

We subsequently analyzed the structural model after verifying the measurement model's validity and reliability adequacy.

### Results of Path Coefficient

The path coefficients of the structural model were analyzed in Table 7. A path coefficient's value is between 1 and −1; a strong positive relationship is represented by a path coefficient approaching 1, whereas a strong negative relationship is represented by a path

coefficient approaching −1. However, a path coefficient's statistical significance is governed by the corresponding standard error, *t* value, and *p* value obtained with a bootstrapping program. In a double-tailed test, *t* values of 1.65, 1.96, and 2.57 signify significance at the 10%, 5%, and 1% levels, respectively.

**Table 7.** Results of path coefficient analysis.

| Research Hypothesis | Hypotheses Relationship | Path Coefficient | Standard Deviation (STDEV) | *t*-Values | *p*-Values & Significance | Supported |
|---|---|---|---|---|---|---|
| H1a: FKG > SMI | + | 0.463 | 0.071 | 6.498 | 0 *** | Yes |
| H1b: IKG > SMI | + | 0.25 | 0.073 | 3.418 | 0.001 *** | Yes |
| H2a: SMI > PS | + | 0.196 | 0.05 | 3.933 | 0 *** | Yes |
| H2b: SMI > PSV | + | 0.379 | 0.048 | 7.934 | 0 *** | Yes |
| H2c: SMI > PB | + | 0.226 | 0.06 | 3.794 | 0 *** | Yes |
| H2d: SMI > PBR | + | 0.27 | 0.046 | 5.86 | 0 *** | Yes |
| H3a: SMI > SMF | + | −0.196 | 0.092 | 2.139 | 0.032 ** | No |
| H3b: SMI > SWB | + | 0.52 | 0.04 | 12.92 | 0 *** | Yes |

Notes: * $p < 0.05$, ** $p < 0.01$, *** $p < 0.001$; FKC = Focused knowledge gain; IKC = Incidental knowledge gain; SMI = Social media intensity; PS = Perceived susceptibility; PSV = Perceived severity; PB = Perceived benefits; PBR = Perceived barriers; SMF = Social media fatigue; SWB = Subjective well-being.

## 6. Conclusions and Implications

### 6.1. Conclusions

This study probed the role played by social media in individuals' perceptions during the COVID-19 pandemic. By applying the S–O–R model, this study identified health belief-related knowledge gain as a driving factor influencing social media intensity and subsequently engendered psychological states, including SWB and social media fatigue.

This study confirmed the significant roles of social media in gaining knowledge to counter-attack the COVID-19 pandemic and further sustain human subjective well-being. Specifically, first of all, both focused and incidental knowledge gain enhanced social media intensity. Second, more substantial social media intensity promoted users' health beliefs regarding perceived susceptibility, severity, benefits, and barriers. Last, social media intensity was beneficial for users' subjective well-being; meanwhile, it was not an issue for social media fatigue.

### 6.2. Theoretical Implications

The findings of this study contributions to related theories can be discussed as follows. First, the present study verified that knowledge gain affects social media intensity. Barker et al. [8] investigated reported knowledge gain from social media from the perspective of social capital affinity. They found that social media intensity enhances incidental and focused knowledge gain, and reported that the relationships of social media intensity with incidental and focused knowledge gain are mediated by the affinity and flow of social capital. By contrast, we postulated that health-related knowledge could actively increase social media intensity during the pandemic. Accordingly, our study confirmed the two-way relationship between social media intensity and knowledge gain and highlighted possibilities related to positive feedback loops.

Second, general studies regarding the effects of social media have focused on only adverse effects, such as social media fatigue [26,37–40], or only positive outcomes, such as SWB [41–44]. Instead, we probed the positive influences of social media and its adverse effects to gain deeper insights into the role of social media during the pandemic.

Third, most studies on social media verified that various overloads caused social media fatigue. However, this study found social media fatigue is not an issue for constructive applications, such as focused or incidental knowledge gain. Namely, knowledge gain would not suffer the problems of overload and fatigue during the pandemic. Moreover,



because social media intensity positively affects SWB, social media is a powerful tool for gaining knowledge related to COVID-19 prevention.

Fourth, this study extends previous research related to the HBM [14,45] by indicating the positive effects social media use has on people's health beliefs.

### 6.3. Managerial Implications

Based on our derived findings, we suggest practical methods for leveraging knowledge to control the spread of COVID-19. First, because knowledge gain is critical for enhancing health beliefs, COVID-19-related information and learning should be designed in a suitable format to facilitate their acceptance and circularity. For instance, Onuora, Torti Obasi, Ezeah, and Gever [46] found that dramatized health messages, such as those presented using animated cartoons, positively affected health behavior in Nigeria.

Second, a host can lead discussions on COVID-19 issues in a social media community to avoid overload and fatigue [47]. Establishing learning communities on social media platforms can improve members' value co-creation behavior, including knowledge sharing [47].

Third, to ensure the correctness and credibility of COVID-19-related information and knowledge, learning communities on social media platforms can implement mechanisms, such as peer reviews and recognition, for different levels of contributions to encourage constructive engagement.

### 6.4. Limitations and Future Research

Despite providing valuable insights, our executed research has certain limitations. First, the data of this study were collected from Taiwanese individuals during the first nationwide lockdown in Taiwan; thus, the derived findings may not be generalizable to different cultural contexts or pandemic stages. Future studies may examine the aspects investigated in this study under different cultural contexts to improve the robustness of the findings. Second, this study was conducted under a cross-sectional design. Future studies can adopt a longitudinal plan for investigation. Future researchers can also investigate the influence of social media use on consumer behavior during the pandemic. Finally, future studies can determine suitable methods for identifying misinformation.

**Author Contributions:** H.-M.C.: Conceptualization, Methodology, Software, Supervision, Writing—Original draft preparation, and Writing—Reviewing and Editing. Y.-D.L.: Conceptualization, Data curation, Visualization, Writing—Original draft preparation, and Writing—Reviewing and Editing. All authors have read and agreed to the published version of the manuscript.

**Funding:** This research received no external funding.

**Institutional Review Board Statement:** Not applicable.

**Informed Consent Statement:** Not applicable.

**Data Availability Statement:** The data presented in this study are available on request from the corresponding author.

**Conflicts of Interest:** The authors declare no conflict of interest.

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
