# Peer review of "Sustaining the Benefits of Social Media on Users’ Health Beliefs Regarding COVID-19 Prevention"

_sustainability, doi:10.3390/su14084809_

Round 1

Reviewer 1 Report

Thank you the opportunity to revise the paper titled: “Sustaining the benefits of social media on users’ health beliefs regarding COVID-19 prevention”

First of all I would like to congratulate the authors for a well written paper, dealing with an interesting topic. I have some comments that I hope the authors find useful:

First, the abstract seems quite long, so I would recommend reducing it to make it more concise and easier therefore to read for potential interested readers.

In the introduction, I would recommend you to more clearly emphasize what is the gap in the literature and how you contribute to fill it. Also, I think it would be convenient to include a clear research question (in the form of question, if possible, ending with a question mark).

Also in the introduction and abstract (and maybe in the title), there should be mention to Taiwan and the context of study, to delimitate clearly what your study does.

Section 3 I would just name it “Research model and hypotheses development” (no need for “present study” and also hypotheses in plural.

When was the questionnaire conducted? What was the language and were all respondents fluent in such language?

Did you check for potential early versus late respondent bias?

There were many cases (and systematic bias) in those who started the questionnaire and didn´t finish it?

I applaud your efforts in checking common method bias, but I think it is better to include common method bias before the results

In the discussion, you summarize your findings and then comment on the practical contributions. However, I would suggest that you describe more how those findings contribute to theory(ies)as well

Good luck with your research!

Author Response

Dear reviewer:
Thank you the opportunity to revise the paper titled: “Sustaining the benefits of social media on users’ health beliefs regarding COVID-19 prevention”

First of all I would like to congratulate the authors for a well written paper, dealing with an interesting topic. I have some comments that I hope the authors find useful:

Response: We appreciate your warm encouragement and support for this study. Our great pleasure is to have helpful feedback from you that drives us to work hard to enhance our paper's quality.

  1. First, the abstract seems quite long, so I would recommend reducing it to make it more concise and easier therefore to read for potential interested readers.

Response:

Thank you for this helpful comment. We rewrote the abstract accordingly to make it more concise and readable.

  1. In the introduction, I would recommend you to more clearly emphasize what is the gap in the literature and how you contribute to fill it. Also, I think it would be convenient to include a clear research question (in the form of question, if possible, ending with a question mark).

Response:

We appreciated this insightful suggestion and tried our best to rewrite the introduction section to emphasize the research gaps we would like to bridge. It is also favorable to demonstrate the research objectives in research questions.

  1. Also in the introduction and abstract (and maybe in the title), there should be mention to Taiwan and the context of study, to delimitate clearly what your study does.

Response:

Thank you for this good comment. Indeed, the paper has to delimitate the research context more clearly. We followed by mentioning it in the abstract and research sample.

  1. Section 3 I would just name it “Research model and hypotheses development” (no need for “present study” and also hypotheses in plural.

Response:

We thank and follow this suggestion.

  1. When was the questionnaire conducted? What was the language and were all respondents fluent in such language?
  2. Did you check for potential early versus late respondent bias?
  3. There were many cases (and systematic bias) in those who started the questionnaire and didn´t finish it?

Response:

We thank the reminder for the vigorous questionnaire survey. We, therefore, described the process in more detail.

  1. I applaud your efforts in checking common method bias, but I think it is better to include common method bias before the results

Response:

Thank you for the notice for us rethinking the meaning of CMB. After referencing the published paper (e.g., Kankanhalli, A., Ye, H., & Teo, H. H. (2015). Comparing Potential and Actual Innovators. Mis Quarterly, 39(3), 667-682.), we moved the section 5.3.1. to 5.2.4.

  1. In the discussion, you summarize your findings and then comment on the practical contributions. However, I would suggest that you describe more how those findings contribute to theory(ies)as well

Response:

We deeply agreed with this excellent comment. We emphasized our findings' contributions in section "6.2. Theoretical implications."

Good luck with your research!

Reviewer 2 Report

  1. Update the introduction section by adding the gaps of the existing studies and the motivation of the work.
  2. It is suggested to add the description of the relationship between the proposed work and sustainability since this journal is a “sustainability” journal.
  3. The motivation towards the proposed research model is not clear.
  4. The authors should add a separate section for Conclusion. In addition, the author should discuss the shortcomings of their proposed work and add future research directions in the conclusion section.
  5. There are some typos errors seen in the manuscript. Authors should proofread the manuscript deeply.

Author Response

Dear Reviewer
Comments and Suggestions for Authors:

  1. Update the introduction section by adding the gaps of the existing studies and the motivation of the work.

Response:

We appreciate your great comment and rewrote the introduction section to highlight the research gaps and our motivation for this study.

  1. It is suggested to add the description of the relationship between the proposed work and sustainability since this journal is a “sustainability” journal.

Response:

Thank you for this meaningful reminder; we intensely treasure the value of sustainability and reflect on it from the research title to research questions and implications.

  1. The motivation towards the proposed research model is not clear.

Response:

We tried our best to respond to this suggestion in the introduction section.

  1. The authors should add a separate section for Conclusion. In addition, the author should discuss the shortcomings of their proposed work and add future research directions in the conclusion section.

Response:

Thank you for this insightful suggestion. We added a separate section (6.1.) for the Conclusion and another section (6.4.) for limitations and future research.

  1. There are some typos errors seen in the manuscript. Authors should proofread the manuscript deeply.

Response:

We did our best to double-check and proofread the manuscript deeply.

Round 2

Reviewer 1 Report

No further comments

Author Response

Summary of responses

Academic Editor Notes:
Please address the final comments
Response:
We appreciate excellent comments from you and concentrated on two primary improvement tracks. First, we double-checked the wording and grammar of the manuscript to ensure meeting the rigorous standards of Sustainability. Second, we made sure the research gaps and our contributions were highlighted clearly. Third, we double- confirmed that our article immersed the vision of Sustainability significantly. For example, we demonstrated social media could be a potent weapon for gaining knowledge to counter-attack the COVID-19 pandemic and further sustain human subjective well-being.

Reviewer1:
No further comments
Response:
Sincerely thank you for your great support.
